# Biomolecular Analysis of Canine Distemper Virus Strains in Two Domestic Ferrets (*Mustela putorius furo*)

**DOI:** 10.3390/vetsci10060375

**Published:** 2023-05-26

**Authors:** Annalisa Guercio, Francesco Mira, Santina Di Bella, Francesca Gucciardi, Antonio Lastra, Giuseppa Purpari, Calogero Castronovo, Melissa Pennisi, Vincenzo Di Marco Lo Presti, Maria Rizzo, Elisabetta Giudice

**Affiliations:** 1Istituto Zooprofilattico Sperimentale della Sicilia “A. Mirri”, Via G. Marinuzzi, 3, 90129 Palermo, Italy; annalisa.guercio@izssicilia.it (A.G.); francesco.mira@studenti.unime.it (F.M.); antonio.lastra@izssicilia.it (A.L.); giuseppa.purpari@izssicilia.it (G.P.); c.castronovo1992@gmail.com (C.C.); vincenzo.dimarco@izssicilia.it (V.D.M.L.P.); 2Department of Veterinary Science, University of Messina, Polo Universitario dell’Annunziata, 98168 Messina, Italy; melissa.pennisi@unime.it (M.P.); maria.rizzo@unime.it (M.R.);

**Keywords:** CDV, ferret, cutaneous lesions, RT-PCR, PCR-RFLP, vaccine

## Abstract

**Simple Summary:**

The current study describes a clinical case of canine distemper virus (CDV) infection in ferrets with a particular focus on viral typing. The diagnostic methods that are described herein allow a rapid and sensitive approach to the diagnosis of CDV infection in a susceptible species. Considering the high morbidity and mortality rates of CDV infection in ferrets and the limited therapeutic possibilities, widespread vaccination remains crucial for preventing the disease and counteracting cross-species infection.

**Abstract:**

Canine distemper is a contagious and severe systemic viral disease that affects domestic and wild carnivores worldwide. In this study, two adult female ferrets (*Mustela putorius furo*) were evaluated for cutaneous lesions. Scab, fur, and swab samples from the external auditory canal, cutaneous lesions, and scrapings were analyzed. Canine distemper virus (CDV)-positive samples underwent RT-PCR/RFLP with the restriction enzyme PsiI, and the hemagglutinin gene sequence was obtained. According to the restriction enzyme and sequence analyses, the viral strains were typed as CDV field strains that are included within the Europe lineage and distinct from those including vaccinal CDV strains. The sequence analysis showed the highest nucleotide identity rates in older Europe lineage CDV strains collected from dogs and a fox in Europe. This study is the first to report on CDV infection in ferrets in southern Italy and contributes to the current knowledge about natural CDV infection in this species. In conclusion, vaccination remains crucial for preventing the disease and counteracting cross-species infection. Molecular biology techniques can enable the monitoring of susceptible wild animals by ensuring the active surveillance of CDV spread.

## 1. Introduction

Canine distemper virus (CDV) is an enveloped virus with a non-segmented, negative sense, single-stranded RNA genome and belongs to the family *Paramyxoviridae*, genus *Morbillivirus* [1]. The CDV genome encodes six structural proteins: nucleocapsid (N), matrix (M), fusion (F), hemagglutinin (H), polymerase (L), and phosphoprotein (P) [2]. Circulating CDV strains cluster into different genogroups according to their geographical distribution and genetic characteristics in particular, according to divergences in the gene sequence encoding the H protein. Based on the nucleotide sequence of the H gene, at least twelve different lineages (America-1, America-2, Arctic-like, Asia-1, Asia-2, Asia-3, Europe Wildlife, Europe/South America-1, South America-2, South America-3, South Africa, and Rockborn-like) have been identified [3]. In recent decades, three CDV genetic lineages have been reported to circulate in Italy: the Europe/South America-1, the Europe Wildlife, and the Arctic-like lineages [4,5,6]. Moreover, studies on the phylogenetic and molecular evolutionary analysis of CDV have revealed that the occurrence of this disease in non-usual hosts is related to mutations affecting the H protein-binding site for the virus receptors [5,7,8].

CDV is the causative agent of canine distemper (CD), an acute, highly severe, contagious, and systemic disease that affects carnivores and other wildlife worldwide [2,9,10,11]. CD is readily transmitted through susceptible hosts via direct contact with oral, respiratory, and ocular fluids, and/or exudates. CDV targets immune cells and, after amplification in lymphoid organs, it spreads through the bloodstream to multiple organs, leading to gastrointestinal, dermatological, respiratory, and neurological signs [11,12,13,14,15]. In particular, CD clinical signs are various and can include dermatitis, hyperkeratosis (also called “hard pad disease”), mucopurulent ocular and nasal discharge with crusts around the eyes and nose, tremors, ataxia, disorientation, paresis or paralysis, and epileptiform convulsions [11]. The main factors that favor the virus’s spread and cross-species infection [16,17] are genetic variability and the broad spectrum of hosts, including members of the *Canidae*, *Procyonidae*, and *Mustelidae*, and the uncontrolled animal movements of stray and domestic dogs [3,18,19,20,21]. The wide and, for some lineages, common CDV host spectrum represents a serious threat to several endangered wildlife species [22,23]. The morbidity and mortality rates of CDV infection may vary greatly among the different animal species. For example, the fatality rate in unvaccinated ferrets is close to 100% [24,25].

Although the presence of CD can be strongly suspected in live or dead animals by observing clinical signs or anatomopathological lesions, respectively, confirmatory laboratory tests are necessary since some clinical signs or lesions could be common to other diseases. These laboratory tests include a broad number of in vivo (using serological and molecular assays) or post-mortem (using histopathology and immunological tissue stains or molecular assays) analyses for a confirmatory diagnosis; however, the specificity and sensitivity of most of these assays are best known for domestic dogs and least known or unknown for most wildlife species. Over recent years, several molecular diagnostic tools have been developed for CDV nucleic acid detection [26], and rapid, sensitive, and specific molecular techniques can differentiate field strains from vaccine strains [26,27,28]. Due to a lack of specific antiviral drugs that are available for therapeutic use against CDV infection in any species, vaccination remains the main measure for disease prevention. Several vaccines against CDV are available for dogs and other domestic and wild animals [29], including subunit, attenuated, inactivated, and DNA vaccines. These vaccines are based on CDV strains belonging to the America-1 (Ondesterpoort, Snyder Hill, Convac, and Lederle strains) and Rockborn-like (Rockborn strains) lineages.

Although CDV infection has been reported in both domestic dogs and wild animals [3,4,5,6] in Italy, and global genomic data on CDV strains collected from ferrets are limited, no sequence for CDV strains from ferrets in southern Italy is available. The present study retrospectively describes two clinical cases of distemper in ferrets reared in Sicily, southern Italy, with a focus on the clinical picture and biomolecular viral typing and characterization.

## 2. Materials and Methods

### 2.1. Case History and Samples Collection

In December 2010, two adult female ferrets were evaluated by veterinary practitioners in Palermo, Sicily (Italy). The two ferrets were purchased two years earlier and were raised in a facility, along with dogs and rabbits, in a rural area within the municipality of Caltavuturo (Palermo province). These animals were used for rabbit hunting, a traditional activity of Sicilian hunters, as a hunting aid to facilitate the capture of rabbits. From 2010 to 2021, this type of hunting was subject to temporal and territorial restrictions according to the local authority of Sicily. Since 2021, rabbit hunting with ferrets has been banned throughout Sicily. The animals were unvaccinated, and information before purchase remained unknown.

The ferrets’ clinical history reported the presence of pyodermitis and pruriginous cutaneous lesions for two weeks that were effectively treated with antibiotics (enrofloxacin). The ferrets were then vaccinated against CDV with a live attenuated vaccine, including a CDV Onderstepoort strain (Eurican^®^, Merial Italia S.p.A., Assago, Italy.). Eight days post-vaccination (eight dpv), the two ferrets were brought to our attention due to severe cutaneous and respiratory clinical signs.

The animals were subjected to clinical examinations, and scab and fur samples, cutaneous scrapings, and swabs from the external auditory canal and cutaneous lesions were collected from both animals and analyzed. Virological, bacteriological, mycological, and parasitological examinations were performed for diagnostic purposes at the Istituto Zooprofilattico Sperimentale della Sicilia “A. Mirri” in Palermo (Italy).

The animals died one week after the clinical examination. The owner telephonically reported the worsening of both respiratory and cutaneous signs, which remained localized to the same areas. A post-mortem examination was not performed.

### 2.2. RNA Extraction

Samples obtained from cutaneous scrapings were homogenized in 10% *w*/*v* of Eagle’s Minimum Essential Medium (EMEM) (Sigma–Aldrich^®^, Milan, Italy) supplemented with 2% fetal bovine serum (FBS) (EuroClone^®^, Milan, Italy) and an antibiotic–antimycotic solution (1000 U/mL of penicillin G sodium salt, 1 mg/mL of streptomycin sulfate, and 2.5 µg/mL of amphotericin B; EuroClone^®^, Milan, Italy). Clarification was performed via centrifugation at 1500× *g* for 10 min at 4 °C. The ear and skin swabs were immersed in the same medium, then centrifuged in the same conditions. Supernatants were stored at −80 °C until use. Total RNA was extracted from 140 µL of the supernatants using the QIAamp^®^ Viral RNA Mini Kit (QIAGEN S.r.l., Milan, Italy), according to the manufacturer’s instructions.

### 2.3. Detection of Canine Distemper Virus RNA

The presence of CDV RNA was confirmed using a primer pair (Table 1), which was proposed by Barret et al. [30], in a 1-step reverse transcriptase (RT)-PCR protocol amplifying a 429 bp fragment of the P gene. RT-PCR was carried out using the AccessQuick™ RT-PCR System (Promega Italia s.r.l., Milan, Italy) in a 25 μL reaction mix consisting of 12.5 μL of Access-Quick™ master mix, 0.1 μL of 40 U/µL RNasin (Promega Italia s.r.l.), 0.5 µL (20 μM) of each primer DMV 1 and DMV 2 (Table 1), 0.5 μL of AMV reverse transcriptase, 8.4 μL of nuclease-free H_2_O, and 2.5 μL of the RNA extract. Reverse transcription and amplification were conducted under the following thermal conditions: 50 °C for 30 for reverse transcription, 95 °C for 15 min, followed by 35 cycles of 94 °C for 45 s, 56 °C for 45 s, 72 °C for 60 s, and a final extension of 72 °C for 10 min. An amount of 10 microliters of each amplicon was analyzed via electrophoresis on a 2% agarose gel supplemented with ethidium bromide, and the bands were visualized with a UV transilluminator.

Using the OneStep RT-PCR Kit (QIAGEN S.r.l.) and the primer set “B” (Table 1), as described by Demeter et al. [31], an RT-PCR assay, followed by a restriction fragment length polymorphism (RFLP) test using the restriction endonuclease PsiI, was performed to discriminate between virulent field and vaccine strains. An amount of 5 μL of each amplicon was digested with 5 U of the restriction endonuclease PsiI (New England BioLabs, Ipswich, MA, USA) in a 20 μL reaction mix consisting of 2 μL of 10× NE Buffer 4 and 12 μL of nuclease-free H_2_O. The restriction profile was determined via electrophoresis on a 2% agarose gel supplemented with ethidium bromide with a 100 bp reference molecular weight ladder.

### 2.4. CDV Sequence and Phylogenetic Analyses

The CDV H gene sequence was obtained from the CDV-positive samples using the primer pair “C” (Table 1), as described by Demeter et al. [31], in a one-step RT-PCR protocol. The complete CDV H gene sequence was amplified in a single-step PCR protocol using the QIAGEN^®^ OneStep RT-PCR Kit (QIAGEN S.r.l.), as previously described in [3]. The amplicons were purified with the Illustra™ GFX™ PCR DNA and Gel Band Purification Kit (GE Healthcare Life Sciences, Amersham, Buckinghamshire, UK) and submitted to BMR Genomics S.r.l. (Padova, Italy) for direct Sanger sequencing with a set of four primers (the B-for/rev and C-for/rev primer pairs; Table 1). According to an overlapping strategy, sequences were assembled and analyzed using BioEdit ver. 7.2.5 software [32]. The prediction of potential N-linked glycosylation sites was carried out using the NetNGlyc 1.0 web-based tool (https://services.healthtech.dtu.dk/service.php?NetNGlyc-1.0 (accessed on 4 February 2023)). Assembled nucleotide sequences were submitted to BLASTn [33] to search for related sequences in public databases.

Phylogenetic analysis based on the full-length H gene sequences was performed with MEGA X software [34], selecting the best-fit model of nucleotide substitution with the lowest Bayesian information criterion (BIC) and using the maximum likelihood method. To elucidate the genetic relationships between the analyzed CDV sequences and other CDV sequences retrieved from public databases, 2 phylogenetic trees (including (i) the CDV sequences of field and vaccine strains of different lineages and (ii) the top 50 CDV sequences using the megablast program selection available in the GenBank database, along with the most recent Europe lineage strains from Italy [35]) were constructed according to the Tamura 3-parameter (T92) model with a discrete Gamma (+G) distribution and invariant sites (+I) (dataset (i): T92+G; dataset (ii): T92+G+I), and bootstrap analyses with 1000 replicates [36]. These sequence data have been submitted to the DDBJ/EMBL/GenBank databases under accession numbers OQ448914 (CDV_Ferret_Italy_2010_IZSSi_90464c2_10) and OQ448915 (CDV_Ferret_Italy_2010_IZSSi_90464c4_10).

### 2.5. Bacteriological, Parasitological, and Mycological Analyses

Bacteriological analysis was performed on samples collected from cutaneous lesions and from the auditory canal, which were placed on blood agar plates and incubated at 37 °C in an aerobic environment for 24–48 h. A mycological exam was conducted on fur and scab samples that were seeded on plates with Sabouraud dextrose agar and cultured at 27 °C for at least 2 weeks. The samples were inoculated directly onto the solid culture media. The identification of bacteria and fungi was realized using biochemical tests and macro- and microscopic observation of the colonies. A parasitological exam was carried out on the samples from cutaneous scrapings using a light microscope at a small magnification (10×).

## 3. Results

### 3.1. Clinical Examination

In the clinical examination, the animals showed cutaneous lesions consisting of pruriginous and squamous-purulent dermatitis on the chin, in the external auditory canal, and in the peri-labial and peri-vulvar regions, with recent (erythema, papules, and vesicles) and old (pustules and dry scabs) lesions and desquamation areas with yellowish-white scales. Breathing difficulties due to nose and muzzle lesions were also observed. CDV infection was considered the main clinical suspect. Sarcoptic mange, primary bacterial infection, fungi or yeasts, cutaneous lymphoma or other neoplasms, pemphigus foliaceus, and distemper were included for differential diagnosis.

### 3.2. CDV Detection and RFLP Analysis

The samples obtained from cutaneous lesions (cutaneous scrapings) tested positive for CDV using the RT-PCR assay. A 429 bp fragment of the P gene was detected. Amplicons remained undigested after the RT-PCR/RFLP analysis and did not generate the 294 and 816 bp fragments that are typical of vaccinal CDV strains, suggesting that the CDV strains from the cutaneous samples of the two ferrets were field strains.

### 3.3. Sequence and Phylogenetic Analysis

From both analyzed samples, the full-length H gene sequence (1824 nts) was obtained (CDV_Ferret_Italy_2010_IZSSi_90464c2_10 and CDV_Ferret_Italy_2010_IZSSi_90464c4_10). The reciprocal comparison of the sequences showed a high identity rate (99.94%) between the two CDV sequences. The comparison of the two sequences showed only one nucleotide substitution (t1645c) that changed the corresponding amino acid (Y549H). The comparison with related sequences in the GenBank database showed the highest nucleotide identity rates with the CDV Europe lineage reference sequences of strains collected from dogs in Denmark in 1991 (98.46–98.19%; accession numbers AF478545, AF478549, AF478543, and Z47761), in Greece in 2013 (98.30–98.19%; JN008899, JN008896, JN008902, and JN008893), and in Germany in 1994/1995 (98.25–98.19%; Z77671 and Z77672). Slightly lower nucleotide identity rates (98.14–98.08%) were observed with the sequences of strains collected from dogs in Italy in 2002/2003 (DQ494317, DQ494318, and DQ494319) and from a fox in Greece in 2005/2007 (JN009807). A total number of eight potential glycosylation sites at the amino acid positions 19–21, 149–151, 309–311, 391–393, 422–424, 456–458, 587–589, and 603–605 were recognized as in other CDV strains of the Europe lineage, which were distinct to those observed in vaccine strains.

The phylogenetic analysis based on the H gene sequences of the field and vaccine strains of different lineages (Figure 1) showed that the analyzed strains clustered within the Europe/South America-1 lineage, separately from those including CDV strains of other lineages observed in Italy over the years (the Europe Wildlife and Arctic-like lineages) or vaccinal CDV strains (included in the America-I and Rockborn-like lineages).

The phylogenetic analysis based on the H gene sequences of the CDV strains from the ferrets and the top 50 CDV sequences from the megablast BLASTn selection, along with the most recent Europe lineage strains from Italy (Figure 2), showed that the analyzed strains clustered within the older Europe lineage strains collected from dogs and foxes in Europe, separately from those clades including the most recent CDV strains collected in northern Italy (including in clades “a” and “b”).

### 3.4. Bacteriological, Parasitological, and Mycological Results

The samples that were collected from the lesions of the two studied ferrets underwent several diagnostic laboratory tests: the bacteriological tests (using swabs from the external auditory canal and cutaneous lesions) allowed the isolation of *Escherichia coli* and *Staphylococcus* spp.; the fungal culture (using scab and fur samples) was positive for dermatophytes; and the microscopic examination of the cutaneous scrapings was negative for ectoparasites.

## 4. Discussion

CDV is a highly contagious viral pathogen that is lethal to both wild and domestic, and land and sea living species of mammals, with high morbidity and mortality rates associated with immunosuppression [24]. According to CDV sequence analyses, which are mainly based on the H gene sequence, three CDV lineages have been reported in Italy: the Europe/South America-1, Europe Wildlife, and Arctic-like lineages [4,5,6]. The first to be detected in both domestic dogs and wild carnivores was the Europe/South America-1 lineage, while the other lineages have been prevalent in wildlife and only sporadically reported in domestic dogs (Europe Wildlife) or, vice versa, reported mainly in Italian dogs and, less frequently, in wildlife (Arctic-like) [4,5,6]. More recently, two different sub-clades within the Europe/South America-1 lineage have been reported in northern Italy: one clade (clade a) originated from older epidemic waves observed both in domestic and wild carnivores, and the other one (clade b) includes CDV strains that were most likely introduced from the Balkans area [35,36,37]. These data indicate the circulation of different CDV strains across the domestic/wildlife interface, with potential threats to the host species and the maintenance of specific lineages in specific ecological niches with the occurrence of temporal epidemic waves. To date, there is limited information on the CDV spread in animals in southern Italy, in both domestic dogs [3,38,39] and wildlife [40], and, to the best of our knowledge, no sequences of CDV strains from ferrets in southern Italy are available. To contribute to the current knowledge on the evolution of the canine distemper virus, this study provided a retrospective molecular characterization of two CDV strains collected from ferrets in southern Italy in 2010 and compared them with currently available CDV sequences.

Our results, based on RFLP, sequence and phylogenetic analyses, and the prediction of potential N-linked glycosylation sites [41,42], suggest that the collected strains must be referred to as field strains of the Europe/South America-1 lineage, and, therefore, we can affirm that the amplified strains were not related to the vaccinal strain used in immunizing prophylaxis and nor to other CDV strains used in other vaccine formulations. In the current outbreak, the origin of the infection remains unknown as no information on other cases of infection in other domestic and wild animals in the same area, including those in close contact, are available. Moreover, because these ferrets were reared in close contact with hunting dogs and used for hunting and were thus potentially in contact with other wildlife host species, a domestic or wild source of infection cannot be ascertained. However, according to the sequence analyses, these strains resulted as being related to CDV strains that have circulated mainly in domestic dogs and wildlife in the past decade, and their closer relationship with the CDV strains from dogs suggests this species was a probable source of infection. In the last decade, the Europe/America-1 lineage continues to be reported in wild animals [35,36,37] along with the Europe Wildlife lineage [43,44,45,46], which has been almost completely replaced with the Arctic lineage in domestic dogs [3,38,44]. Most of these studies are limited to the northern/central regions of Italy, except for a study in domestic dogs [3], indicating the need to carry out further epidemiological evaluations of the wildlife in southern Italy in addition to the current limited studies [40,47].

The cutaneous lesions were suggestive of CDV infection, having been previously observed in natural infection [48,49]. The possibility of the lesions being attributable to primary bacterial or fungal infections was not excluded, although this occurrence was considered unlikely. More probably, but not certainly, the bacteria and fungi that were isolated from the samples collected from the lesions could be linked to the immunosuppression caused by the virus, which is particularly severe in ferrets [50,51,52,53]. From a clinical point of view, the prevalently dermatological picture observed in the animals under study differs somewhat from the classic signs described in ferrets, where respiratory and neurological involvement is common.

It was also considered that the clinical course (4 weeks) in the studied ferrets could have been mitigated with a previous hypothetical vaccination, as in the case described by Zehnder and colleagues [54], which observed similar dermatological lesions and a prolonged clinical course (3 weeks) with the absence of respiratory and neurologic signs in the vaccinated ferrets. Therefore, they assumed that the clinical signs might have been partially attenuated by the immune coverage provided by the previous vaccinations or by an unidentified reduced-virulence strain. Nonetheless, Zehnder and colleagues [54] highlighted that CDV should remain a clinical suspicion for ferrets with cutaneous lesions with low responsiveness to treatment, even in previously vaccinated animals. Limitations due to the lack of complete information, post-mortem examinations, and histological/immunohistochemical assays prevented further descriptions and conclusions.

Variations in the H gene sequence have been related to host species shift or the virulence of CDV strains [55,56,57]. Variations in specific amino acid residues of the SLAM-binding region, particularly at the 530 and 549 residues, have been associated with distemper adaptation in non-dog hosts [58]. The amino acid substitution of tyrosine (Y) with histidine (H) at site 549 has been associated with an adaptation of CDV from dogs to non-domestic hosts [59]. In the sequences of the two ferrets, glycine (G) at site 530 was common to CDV strains from dog and non-domestic hosts, while at site 549, only one divergence between the two strains (Y/H) was observed, preventing us from observing any evidence of adaptation to non-canid hosts or virulence. As evidenced in other studies, non-canid hosts are just as likely to be infected with CDV strains with H-549 Y or H as canids; however, canids are more likely to be infected with CDV strains with 549Y [59].

The vaccines in current use are safe, and their efficacy has contributed to the reduction in CDV infection in domestic dogs, preventing spillovers of the disease into non-domestic hosts [2,59]. Clinical cases of distemper after vaccine administration have been reported in both domestic and wild carnivores [60,61] and have been related to virulent field strains rather than vaccinal reversion. Evidence of vaccinal CDV strains in clinical samples has also been reported [28,44], highlighting the need for clear discrimination between field and vaccine strains, particularly if vaccines have been administered a few days before the onset of clinical signs. In the studied cases, both ferrets received a vaccinal dose, including a CDV Onderstepoort strain, a few days before the diagnostic assays, and the analyzed strains have been related to field strains, excluding the causal evidence of vaccinal strain shedding or any potential residual virulence.

## 5. Conclusions

Despite the limitations due to the lack of autoptic examinations or additional histochemical assays, we can assume, based on the results hitherto shown, that the cause of death of the ferrets was likely linked to the infection caused by a virulent field canine distemper strain, which was probably followed by, or occurred in co-infection with, bacterial and fungal microorganisms. The current study enriches the bibliography on clinical cases of natural CDV infection in mustelids while implementing the available data on viral field strains. Active surveillance of spreading CDV strains using molecular biology techniques can help to monitor susceptible wild animal hosts, particularly endangered species, thus supporting global animal welfare. Given the virulence of CDV in ferrets and the limited therapeutic approaches, vaccination is crucial for the prevention of this disease in this “emergent pet”, considering the various potential sources of infection [62]. Further studies in non-domestic hosts and, particularly, in wildlife in southern Italy are recommended.

## Figures and Tables

**Figure 1 vetsci-10-00375-f001:**
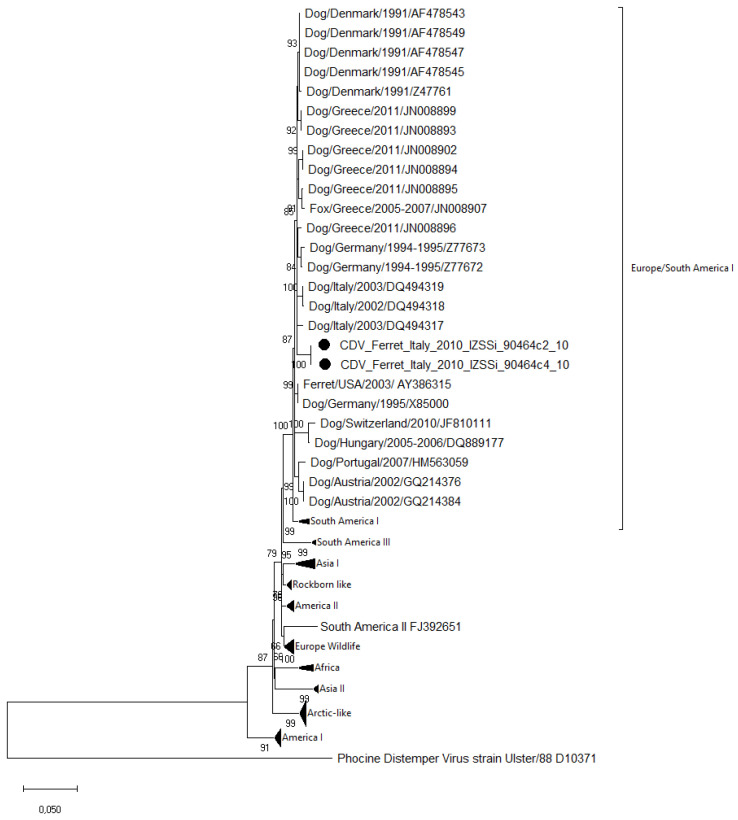
Maximum likelihood phylogenetic tree based on the full-length hemagglutinin (H) gene sequences of the CDV strains, indicated with black dot markings, identified in this study, and 97 CDV sequences available in the GenBank (1000 bootstrap replicates; bootstrap values greater than 65% are shown). CDV Europe lineage strains are indicated with host species/country of collection/year of collection or submission/accession number. CDV sequences of other lineages with the following accession numbers in parentheses are grouped and collapsed: South America I (FJ392652, JN215474, and JN215476); South America III (KF835420, KF835414, and KF835425); Asia I (AB212965, AB329581, D85754, FJ409464, HQ540293, and AF178038); Rockborn-like (AF178039, GU266280, FJ461702, AY964114, GU810819, and FJ705238); America II (Z47762, Z47764, AY526496, Z47763, Z47765, AY498692, and AF164967); South America II (FJ392651), Europe Wildlife (DQ889188, DQ889187, Z47759, DQ228166, GQ214374, GQ214369, JN153021, JN153022, and JN153023); Africa (FJ461714, FJ461698, FJ461720, and FJ461724); Asia II (AB040767, AB040768, EU252149, AB025270, and AB252718); America I (AF305419, AF014953, DQ903854, AF378705, Z35493, AM903376, EF418782, DQ778941, AY548109, AY466011, and AF259552). Phocine distemper virus sequence (strain Ulster/88; D10371) was used as an outgroup.

**Figure 2 vetsci-10-00375-f002:**
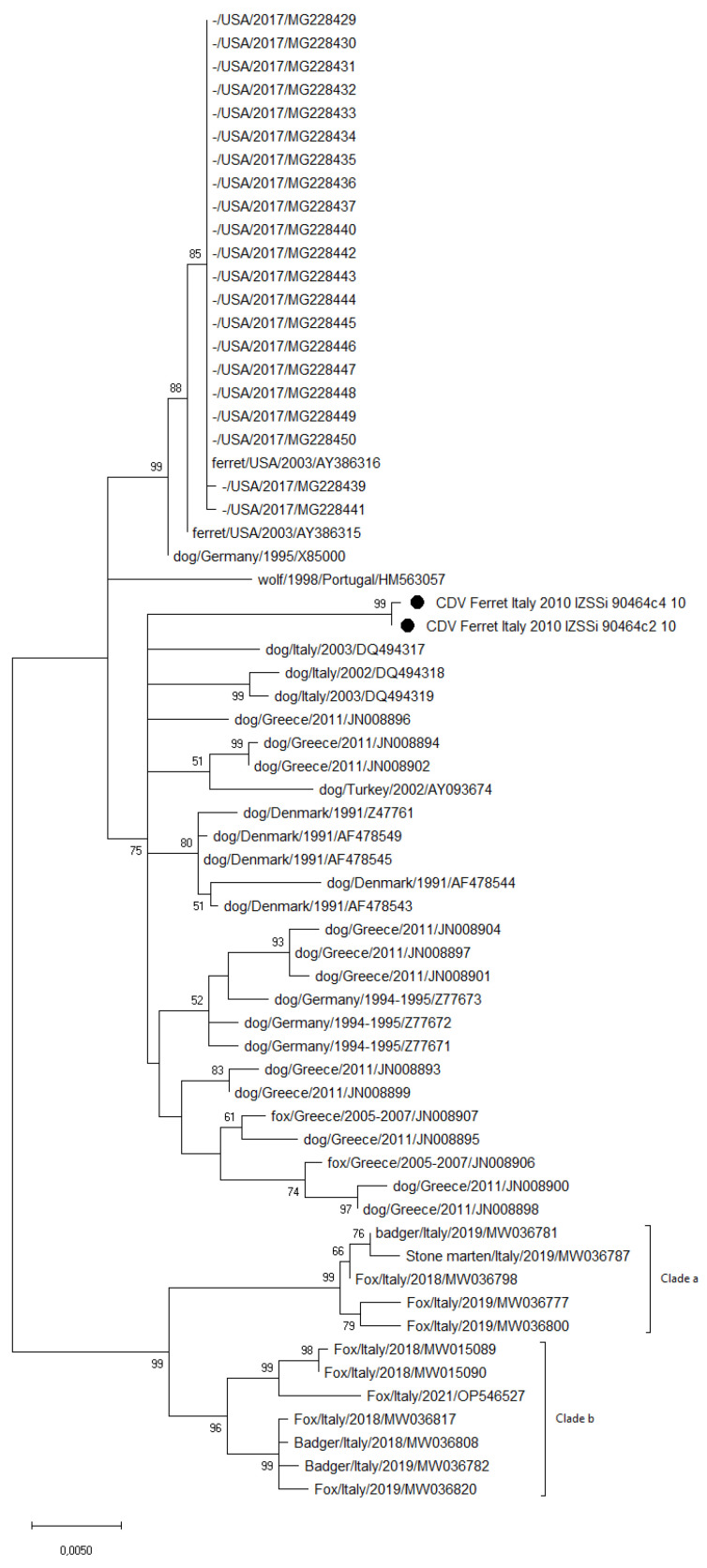
Maximum likelihood phylogenetic tree based on the full-length hemagglutinin (H) gene sequences of the CDV strains, indicated with black dot markings, identified in this study and 62 CDV sequences of the Europe lineage available in the GenBank (1000 bootstrap replicates; bootstrap values greater than 50% are shown). CDV strains are indicated with host species (“-” when unavailable)/country of collection/year of collection or submission/accession number. The most recent Italian CDV sequences from non-domestic hosts are grouped in square brackets under the clades “a” and “b”.

**Table 1 vetsci-10-00375-t001:** Oligonucleotide primers used in the biomolecular assays.

Method	Primer Sequence (5′-3′)	Target	Size	Reference
RT-PCR	DMV 1: ATG TTT ATG ATC ACA GCG GT’ DMV 2: ATT GGG TTG CAC CAC TTG TC	P gene	429 bp	[30]
RT-PCR/RFLP	B-for: AGG CCG TAC ATC ACC AAG TC B-rev: TGG TAA GCC ATC CGG AGT TC	H gene	1110 bp	[31]
Sequence analysis	C-for: AAC TTA GGG CTC AGG TAG TC C-rev: AGA TGG ACC TCA GGG TAT AG	H gene	2023 bp	[31]

## Data Availability

The sequences obtained in this study have been submitted to GenBank under accession numbers OQ448914 (CDV_Ferret_Italy_2010_IZSSi_90464c2_10) and OQ448915 (CDV_Ferret_Italy_2010_IZSSi_90464c4_10).

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
