# Peer review of "Biomolecular Analysis of Canine Distemper Virus Strains in Two Domestic Ferrets (Mustela putorius furo)"

_vetsci, 2023, doi:10.3390/vetsci10060375_

Round 1

Reviewer 1 Report (Previous Reviewer 1)

-   The Authors have revised the manuscript by integrating and correcting the points noted in the first instance, in particular by emphasizing, as suggested, the scientific contribute that appeared to be the most innovative, in other words the typing of new CDV strains in ferret.

-       Here are some specific minor revisions:

-       - Line 30: I suggest ending the sentence by specifying what the samples were analyzed for (for example….”subjected to bacteriological, virological and parasitological tests”);

-       - Line 42: the ferret, understood as Mustela putorius furo, is not a wild animal (delete wild) but a domestic animal selected from wild Mustela putorius.

-       - Lines 107-112: the clinical aspects observed by the Authors are results (unlike the clinical aspects collected with the anamnesis, which can actually be included in the Materials and Methods): I would suggest moving this part to the Results, in a separate sub-chapter "clinical examination", recovering the deleted photos (useful for showing the lesions described).

Author Response

Reviewer 2 Report (Previous Reviewer 2)

The manuscript summarise a case and publish  CDV  sequences from two domestic ferrets.

The main weakness of the case report is the retrospective point of view, the lack of the direct pathological examinations of the cadavers followed by thorough histopathological and microbiological examinations. The manuscript, however, reports all of the feasible examinations and their results within the objective limitations. 

Nevertheless, here are some major points for further improvement of the manuscript:

- Use the same sample type names referring to the same sample to make clear the whole sample process, e.g. in line 115-116 the samples are listed, including 'cutaneous scrapings', but the 'samples obtained from scarification of cutaneous lesions' is mentioned in line 123. If you mean the same, name it same! Please, check it throughout the text.

- In chapter 2.4 prediction of potential glycosylation sites is mentioned, but I miss its results and the discussion. 

-Chapter 2.5 contains not any reference. It is not wrong, but then you should describe the details of the process, like suspension, dilution, plating etc.

- Chapter 3.1 What do you mean? According to the RFLP the strain is not a vaccine strain? It is not clear from the text.

- E. coli and Staphylococcus: Clarify, whether the dermatology lesions were symptomatic for these bacterial infection. If yes, there can be considered as secondary infections, if not, they should not be considered as causative agents. Support it by references. Discuss it in the 'Discussion', modify the Conclusions, if needed.

- Intensive screening and improvement in English usage is needed. Not only to correct some inadequate forms ( 'underwent to'; 'referred to as'; 'center Italy' 'telephonically reported'; 'amplification were conducted under' etc.), but to make the whole text concise and straightforward.

- The 2nd paragraph of Discussion is totally redundant, cut it!

Minor points:

- in the 2nd line of Discussion: 'different animal host species' -too general, specify! (Mammals? Carnivors?) 

- Delete the typed hyphens from divided words, they have moved during formating.

- The first mention of CDV is in the 4th line of Abstract, not the 7-8, use the abbreviation in the second case.

- line 105: Eight days post vaccination (8 dpv)

Author Response

This manuscript is a resubmission of an earlier submission. The following is a list of the peer review reports and author responses from that submission.

Round 1

Reviewer 1 Report

The Authors report a case of suspected distemper in two domestic ferrets. The typing of a new CDV strain in Mustela putorius furo represents the strength of the article and the greatest contribution to the knowledge of the infection in this animal. The weakness, as specified in the comments below, is the lack of autopsy findings, which could give evidence (by histopathology and PCR) about both the distribution of the wild strain in organs and tissues and the possible pathogenic action of the vaccine strain used; in addition to this, there are some inconsistencies (they have to be clarified in the eventual revision)  relating to the Materials / Methods (inside this chapter there are various parts concerning Results or Discussion) as well as to the description and interpretation of the Results (for last which the specific chapter is missing).

Comments

Line 2: in title, change "strains" in “strain” as only one field CDV have been detected.

Line 25: as for RFLP-PCR in line 26, correct in “Real time-Polymerase chain reaction (RT-PCR)”

Line 30 and 128: it would be more suitable to speak about “nucleotide similarity” and not homology.

Line 31 and line 129 and line 175: the fox strains with nucleotide similarity to FURETTO CDV strain come from Germany and not Italy as shows the phylogenetic tree in Figure 3; check the data.

Line 85-86: eliminate these lines (probably an informal communication between the Authors)

Line 87; lines 178-180; lines 195-196: some confusion about the origin of the two ferrets; explain exactly where the ferrets came from and lived. As the ferret are domestic and have right been defined domestic in the title, why or how these animals were “hunted” (probably captured alive?) in the Madonie area ? Or purchased as adults as written in lines 195-196? However insert the anamnesis in Material and Methods and not in Discussion.

Lines 92-96 (with the Figures 1 and 2): these are Results (of clinical examination) and not Material and Methods. Note: Results Chapter kis missing, it have to be created.

Line 102: the figure about perilabial lesions is missing

Line 106: here and further on in results (especially in line 137 for microbiological tests, line 140 for PCR results, samples Lanes 1-4 in legend of Figure 4 and everywhere necessary) the Author have to express if the results are referred to one or both ferrets…the new CDV strain was detected in both animals?

Lines 124-125 until Europe lineage: these are Results and not Material and Methods.

Lines 125-129 and lines 142-146: : these are Discussion and not Material and Methods or Results.

Line 107: It would be useful insert a short note about methods of bacteriologic, mycologic and parasitological exams (culture media, T° and atmosphere of incubation, solutions, etc.)

Line 140: RT-PCR?

Figure 4: insert the bp reported in the legend also to the right side of the image in order to make evident the correspondence with the bands.

Line 217 and following: as mentioned, the lack of autopsy make difficult establish what exactly was happened. But after considering various bibliographic references and the results obtained in the study, which is the hypothesis of the Author about their case? In other words, wild virus or vaccine strain or the two virus together could be responsible of the death of both ferrets?

Reviewer 2 Report

The article reports a recent CDV infection in domestic ferrets. Since it is a 'case report', there are limited expectations to the reported results. Frankly, the one and only relevant information of the study is the sequence data of the virus, however it is poorly demonstrated. Line 142-143: “The sequences are part of a well-defined cluster of strains belonging to the lineage Europa.” But it isn't demonstrated in Figure 3, there is not any cluster signed. There is a possibility also in software package MEGA 5 to sign and name the clades (America-1 & 2, Europe, Wildlife, etc.) within the tree. There are contradictions between Table 2 and Fig. 3, e.g. position of sequence AY964114.

The sequence should be uploaded to NCBI/EMBL database (Genbank), the Accession Number. After it you can label the new sequence like the others, instead of calling it ‘FURETTO’. In caption should indicate that ‘diamond represents…’.

Classification of the sequences should be supported by reference. The reference of the classification system (‘according to …’) should be indicated in Figure 3 and Table 2 as well as in the text (M&M and/or Results). It should be also explained why this nomenclature was chosen. 

In the middle section of the manuscript there are plenty of fundamental editing mistakes, the M&M, Results, Discussion sections should be clearly separated, e.g.:

-        there is no “Results” head

-        Lines 85-86 – some blank text remained

-        Lines 125-129 belongs to Results, should paste in the end of results

-        Lines 134-137 Materials & Methods, partly redundant info

-        Lines 173-175 Results

Discussion section after Line 200 is about irrelevant topics, thoughts are not supported by results, they become rather hypothetical. Since it is a case report, this part should be reduced, remaining very close to the results.

Figures and Tables should be captioned to be understood ‘standing alone’, without reading the text.

Result of RFLP and Fig. 4 is out of context. Its necessity is not explained, even after presenting the sequence and phylogenetic position of the virus. In Fig. 4 only one pattern is represented (Bussel strain), neither other possible patterns nor any known strains with the same pattern. The RFLP should be repeated with more other strains or at least (the lack of) the recognition sites for the restriction enzymes should be presented by the simplified alignment of the different strains in an additional figure.

English language should be improved, there are several examples of strange usage of words and phrases (Line 27: ‘segregated into’; Line 186: ‘According to our results’ etc.). Critical overview is needed!!!

Minor mistakes

-        Restriction fragment length polymorphism - Polymerase chain reaction: Start the words capitals or lower cases, not mix them. Use the form PCR-RFLP, since it is the order of the process.

-        Line 110: in Table 1 not ‘protocols’ are listed, only the primer pairs.

-        Table 1: include references of primers

-        Line 138: use Italics for bacterial names

-        Table 2: Hungary, not Ungheria

-        Line 125: Europe, not Europa

Reviewer 3 Report

The manuscript “Biomolecular analysis of canine distemper virus strains in domestic ferrets (Mustela putorius furo)” is an interesting report of a clinical case of distemper in ferrets, from which CDV strains were genetically characterized. Unfortunately, the paper needs to be improved on different levels before considering it suitable for publication. English writing, wording and grammar need to be checked and revised, since many sentences are confusing, and the meaning is often unclear. The structure of the paper is inconsistent, Materials and methods section contains details on the results and the Results section is lacking. The structure should be revised, appropriate sections should be added, or sections should be removed and content completely reorganized, if suitable with the journal guidelines.

Major:

There is no clear and organized presentation of the clinical case and anamnesis of the animals. Information should be added such as the type of animals (pets from private owner of from a breeder, animals used for hunting???), exposure to other animals, the indication of where the animals were visited (vet practice, clinic, university veterinary hospital, …), if the case was referred, why sick animals were vaccinated, …

The clinical case should be presented completely before presenting and discussing the results, including the fatal outcome of the infection.

Materials and methods section is merged with Results section which is not present in the manuscript and should be added grouping the results of all the analyses.

The description of methods and results of the phylogenetic analysis is unclear (not all American strains appear to be vaccines from the authors’ selection of reference strains, but line 126 is confusing; ) and there is no separation between methods and results.

In the Discussion, the authors present a lot of bibliography in detail losing focus on their results, the discussion should be reorganized and present the elements in order with a briefer mention of the respective references. The fact that vaccination was performed on animals with symptoms anyway should be addressed, motivated and discussed.

Minor:

CD virus should be written as CDV.

Lines 85-86 should be deleted.

Modify line 91 “Despite vaccination”, which should not be intended as a therapy attempt.

Line 107: “bacteriologic, virologic, mycologic and parasitological examinations” should be detailed and results should be stated, even if negative.

Table 1 should contain the references for each method, and it is not clear what “phylogenetic analysis” means in comparison to the previous two tests.

Lines 117-118 are not clear, is the obtained sequence 1823 nucleotide long?

Check wording in lines 124 (analysis was resulted positive)

Line 120: replace “isolated” with “collected” if not all reference sequences were viral isolates.

Line 125: the origin of the strain could be from Europe or the strain could belong to a lineage…revise the sentence please

Line 128: are the Italian references all “isolates”?

Line 134: replace “they” with “the animals” or “the ferrets”

Line 148: delete “strips”

In line 186, the sentence starts with “according to our results” and then only literature is described, so the meaning is unclear.

Line 218: rephrase the sentence please, it seems that natural disease is linked to vaccination.